# The Project “Colourful Means Healthy” as an Educational Measure for the Prevention of Diet-Related Diseases: Investigating the Impact of Nutrition Education for School-Aged Children on Their Nutritional Knowledge

**DOI:** 10.3390/ijerph192013307

**Published:** 2022-10-15

**Authors:** Elżbieta Szczepańska, Agnieszka Bielaszka, Agata Kiciak, Gabriela Wanat-Kańtoch, Wiktoria Staśkiewicz, Agnieszka Białek-Dratwa, Marek Kardas

**Affiliations:** 1Department of Human Nutrition, Department of Dietetics, Faculty of Health Sciences in Bytom, Medical University of Silesia in Katowice, 41-808 Katowice, Poland; 2Department of Technology and Food Quality Evaluation, Department of Dietetics, Faculty of Health Sciences in Bytom, Medical University of Silesia in Katowice, 40-000 Katowice, Poland; 3Department of Health Promotion, Faculty of Health Sciences in Bytom, Medical University of Silesia in Katowice, 40-000 Katowice, Poland

**Keywords:** nutrition education, children, children’s diet, diet-related diseases, healthy eating

## Abstract

Methods: An educational project called Cooking and Educational Workshops “Colourful means healthy” was conducted at the Department of Dietetics of the Faculty of Public Health in Bytom of the Medical University of Silesia in Katowice, Poland, between 1 July 2017 and 30 June 2019. The participants/recipients of the project were second-grade primary schoolchildren (317 pupils aged 7–9 years). Objective: The aim of this article is to assess the change in knowledge of the principles of healthy eating among children aged 7–9 years following the nutritional education we carried out as part of the “Colourful means healthy” project. As part of project evaluation, the participating children were asked to rate selected food products in terms of their influence on human health (healthy vs. unhealthy). Results: There was a statistically significant difference between the percentage of correct answers provided by the pupils before and after nutrition education. Thus, one may conclude that conducting an educational programme caused the participants’ nutrition knowledge to increase. Conclusion: The present study demonstrated the potential of nutrition education in the form of cooking and educational workshops in terms of increasing nutrition knowledge. As such, workshops like these can be a useful measure for improving eating habits and eliminating dietary errors in the study population. However, future research is needed in order to verify whether such cooking and educational workshops can produce beneficial and lasting changes in dietary habits over the long term.

## 1. Introduction

There is now an ever-growing epidemic of obesity, occurring among children, adolescents and adults, most commonly associated with a positive energy balance as a consequence of excessive energy intake from food and/or insufficient physical activity [1,2]. According to a World Health Organisation (WHO) report, in 2016, overweight and obesity was prevalent in 42 million children under 5 years of age and 340 million aged 5–19 years [2]. In 2015, among the 74 million school-aged children living in European Union countries, 12–16 million were overweight, of which 3–4.5 million were obese [3]. According to observations by researchers at the National Centre for Nutrition Education (NCEŻ) in Poland (formerly the Institute of Food and Nutrition), as many as 22% per cent of Polish school-aged children are overweight, a 12% increase compared to the 1970s [4]. According to a report published in 2018 by the Joint Research Centre of the European Commission, as many as 31–35% of Polish 11-year-old boys are diagnosed as obese or overweight, figures that are worse only in Greece and Cyprus [5].

The data on such a widespread prevalence of overweight and obesity among children are extremely worrying. It has long been known that overweight and obesity cause many diseases in adults, including cardiovascular diseases, type 2 diabetes and malignancies [6,7]. It is also increasingly reported that they cause many serious disorders even in childhood [5,8,9]. These are not only physical complications, which can affect every system in the body, but also psychosocial ones [8,9]. Moreover, many obese adults were obese children in the past [10]. The risk of excessive body weight and associated complications increases with the child’s age and the degree of obesity [8]. According to the NCEZ, children who are successfully restored to a normal weight by the age of 7–13 years do not have an increased risk of developing type 2 diabetes in adulthood, whereas for children who reach a normal weight by the age of 13–18 years, this risk is more than three times higher than in the standard population [11].

Research findings indicate that it is possible to change eating habits to more health-promoting ones by increasing knowledge about proper nutrition. Therefore, it is important to conduct nutrition education programmes [12,13]. Such management is particularly important for young children, as this is the age at which food habits and preferences are formed and often remain unchanged throughout life [14]. Many of the abnormal eating habits of children are related to the diet of their families [15,16]. It is often the case that children only encounter a different, correct way of eating for the first time in their lives when they start school.

Nutrition education programmes targeting the school population are conducted worldwide [17,18,19,20]. In Poland, all schools are obliged to conduct nutrition education. The education programme for grades I-III, in force since 2017, states: “*The task of the school is (...) to consolidate the knowledge of proper nutrition”* [21]. However, as pointed out in the summary of the National Nutrition Congress (Poland, 2019), there is still not enough nutrition education in schools, media and workplaces in Poland, and the reformulation of food products (to reduce their sugar, fat and salt content) is progressing very slowly, despite the regulations being implemented. In Poland, nutrition education is inadequate, there is insufficient funding and there is a lack of early diagnosis and prevention of overweight in schools, among others [22]. This fact, among others, prompted us to develop and implement the educational project “Colourful means healthy” described below and to present the results of nutrition education on knowledge among children aged 7–9 years.

The main objective of our project was to conduct nutrition education classes. Our project aimed to develop pupils’ competencies in communication, cooperation, critical and creative thinking, problem-solving and innovative action. To achieve this goal, didactic activities were planned and implemented to increase interest in the topics of good nutrition and healthy lifestyles. These activities aimed to:−encourage the consumption of products recommended for daily nutrition, based on the latest dietary recommendations for children (communication);−overcome resistance to new (hitherto unfamiliar) foods and encourage their consumption (arousing curiosity);−encourage pupils to get to know new tastes and flavours, fostering acceptance of hitherto disliked foods (innovative measure);−promote the consumption of natural/unprocessed foods (problem-solving and decision-making skills);,−develop the ability to diversify and compose tasty and healthy meals (fostering creativity);−prevent the consumption of excessive amounts of sugar and salt (developing critical thinking).

The aim of this article is to assess the change in knowledge of the principles of healthy eating among children aged 7–9 years following the nutritional education we carried out as part of the “Colourful means healthy” project.

## 2. Materials and Methods

### 2.1. Project Participants

The educational project entitled Culinary–educational workshops “Colourful means healthy” was implemented at the Department of Dietetics of the Faculty of Health Sciences in Bytom, Silesian Medical University in Katowice, between 1 July 2017 and 30 June 2019. The project partner was the city of Siemianowice Śląskie. The participants in the project were all children attending their second year of primary school, i.e., 7–9 years of age, in Siemianowice Śląskie. Every age is appropriate for nutrition education; however, the nutrition education programme must be tailored to the audience. The educational content introduced concerned the principles of rational nutrition; it was consistent with the core curriculum implemented in schools, and provided excellent support for the teachers’ work.

### 2.2. The Inclusion Criteria for Participants in the Study

The criteria for inclusion of participants in the study in the project were: status as a grade II pupil of a primary school in Siemianowice Śląskie, written consent from parents or legal guardians to participate in the study, and participation in all stages of the project. The criterion for exclusion from the study in the project was the absence of the child from one of the nutrition education modules.

### 2.3. Nutrition Education Project

Due to the specific nature of the educational project and the resulting practical nature of the activities, where the highest effectiveness is achieved when working in small groups, it was decided that one group should consist of a maximum of 25 people, with one tutor supervising a maximum of six students. On the other hand, some of the classes were conducted in “master–student” mode, meaning that there was one student per tutor. Research and teaching staff with the relevant qualifications and teaching experience were involved in the implementation of the classes. The methodological details of the programme were developed based on available and up-to-date recommendations on rational nutrition for children [23,24,25,26]. Classes were implemented in the sensory laboratory and technology laboratories of the Dietetics Department of the Silesian Medical University in Katowice.

As part of the project, two cycles of workshops were planned and conducted (each cycle lasted 12 months). A total of 317 pupils aged 7–9 engaged in the educational activities, including 149 girls (47.0%) and 168 boys (53.0%). The project was divided into four original didactic modules: an educational workshop (module 1), a sensory workshop (module 2), a cooking workshop (module 3), and a molecular gastronomy workshop (module 4). The different modules were implemented in two stages. The first stage consisted of modules 1 and 2, and the second one of modules 3 and 4. A one-week break was planned between the stages to consolidate the educational content and to carry out an evaluation. During the break, teaching activities were carried out to allow students to consolidate the content taught previously. These activities were carried out by teachers at the school during daily classes, using teaching guides previously developed by the university staff. This course of action was intentional, as it has a high degree of effectiveness, as shown by many experiences detailed in scientific reports [27,28].

The whole “Colourful means healthy” project was based on the principles of healthy eating currently in force in Poland for children and young people. These principles include: Eat 5 meals regularly and remember to drink water frequently and brush your teeth after eating.Eat a variety of fruit and vegetables as often and as much as possible. Eat cereal products, especially whole grains.Drink at least 3–4 glasses of milk a day (you can replace it with natural yoghurt, kefir and-partly-cheese).Eat lean meat, fish, eggs, pulses and choose vegetable fats over animal fats.Do not consume sugary drinks and sweets (replace them with fruit and nuts).Don’t add salt to your food, eat salty snacks or fast food.Be physically active at least one hour a day (limit TV watching, computer use and other electronic devices to 2 h).Get enough sleep so that your brain can rest.Check your height and weight regularly.

As part of our project, we used the current food pyramid for children and young people aged 4–18 years: The Healthy Eating and Lifestyle Pyramid created by the National Centre for Nutrition Education [26].

A detailed factual description of the project as it was carried out, including the objectives of the individual learning activities and how they were to be conducted, is included in Table 1 and Table 2.

As part of module 1 (Figure 1), students were introduced to the latest Healthy Eating and Physical Activity Pyramid for children and young people in Poland. Participating pupils learned who developed the Healthy Eating and Physical Activity Pyramid, for what purpose, and at whom it is aimed. In addition, they learned about the different levels of the pyramid, including the products recommended and contraindicated in daily nutrition. Attention was also drawn to the harmfulness of salt and the need to replace it with natural spices and the need to hydrate the body. All these activities helped to improve critical thinking skills. Group activities were carried out. During the group activities, pupils solved crosswords and puzzles, e.g., about rare fruit and vegetables or the principles of good nutrition; carried out experiments, e.g., using a non-Newtonian fluid; and participated in the game Familiada. All games aimed at improving creative thinking competencies as well as communication and cooperation skills. An original educational teaching game was conducted. While participating in the educational game, pupils repeated the knowledge acquired in the first part of the workshop, expanding it with new facts. At the same time, being participants in the game, they enhanced their competencies in communication and cooperation in a group.

Module 2 introduced the role that the different senses play in the evaluation of food products. The participating students learned what sense organs are and the role they play in the evaluation of food products. In addition, they were introduced to the basic principles of diversifying and composing meals in terms of their sensory qualities. All these activities helped to improve creative thinking competencies. The ability to recognise basic tastes, odours and colour intensity was assessed. In addition, individual taste sensitivity (taste sensitivity threshold) was assessed and an odour recognition test was carried out. The pupils recognised basic tastes as well as new tastes and odours, which should greatly promote the acceptance of new or previously disliked foods. The principles of evaluating the quality of food products were presented using the example of yoghurt and applying the ranking method. Pupils identified their consumer preferences and assessed the quality of yoghurts using their own senses.

Module 3 of the course included a cooking workshop during which the principles of healthy eating were recalled, paying particular attention to preventing the consumption of excessive amounts of salt and sugar by using natural substitutes. Pupils were introduced to modern equipment and appliances in the technology workshop (Figure 1). The principles of hygiene and safe work in the technology workshop were presented. Culinary classes were conducted in groups, during which students recognised little-known vegetables; learnt the taste and smell of fresh and dried herbs; prepared dishes using, among others, vegetables and fruit; used salt and sugar substitutes to season prepared dishes; and set the table.

The following dishes were prepared during the class: colourful sandwiches consisting of vegetables, whole-grain bread and vegetable pastes; sugar-free oatcakes; vegetable salad with grilled chicken with herbs; pineapple and peach smoothie; green fruit drink with kale; cheese paste with vegetables; sugar-free compote with clove and apricot; and sugar-free fruit jelly with natural yoghurt and fruit.

The final nutritional education module was Module 4: molecular gastronomy workshop (Module 4). In early childhood nutrition education, techniques such as experiential learning and observation are used. In module 4, we used the method of observation, but also curiosity. In this part of the workshop, the children prepared healthy dishes using molecular gastronomy methods and techniques together with the workshop leader. Among others, ice cream was prepared from fruit alone using liquid nitrogen; caviar was prepared from chokeberries using the spherification method, adding calcium to chokeberry juice and alginate drip; and spaghetti was formed from carrot juice using agar-agar and special tubes as moulds.

During the molecular gastronomy workshop, the physical properties of liquid nitrogen and how it can be used in freezing at very low temperatures were demonstrated. The participating students had the opportunity to observe how to prepare and taste ice cream with unusual flavours. In addition, with the help of the instructor, they were able to prepare ice cream on a stick according to their ideas. The effects of very low temperatures on the structure of exemplary foods such as olive oil, fresh herbs and spinach were discussed, allowing the children’s imagination to be stimulated as to how they could be used as salad ingredients.

During the implementation of all classes, didactic methods characterised by the highest effectiveness were used, including giving methods (story, description, demonstration, observation), searching/activating methods (didactic games: brainstorming, situation method, simulation method) and practical methods (individual exercises, group exercises, tests). Throughout the teaching process, the following teaching aids and resources were used to make the content conveyed more comprehensible, to facilitate the thought processes and to help students perform the exercises: a computer; a multimedia projector; a multimedia whiteboard; an original didactic game “Go for Health”; food products such as vegetables, fruit, whole-grain products, lean dairy products, natural yoghurts, and herbs and spices without added salt and sugar; catering equipment and devices; sample sets for sensory tests; and office materials.

### 2.4. The Evaluation Process for Learning Activities

As part of the evaluation of the project, the participating children were asked to evaluate selected foods in terms of their impact on human health (healthy and unhealthy). The first part of the evaluation took place before the start of the project, while the second part took place after the end of the project. The pupils were invited into a room where the following foodstuffs were laid out on tables: water, one-day carrot juice, fizzy drink, breadsticks, chocolate wafer, fruit candy, crisps, salted nuts, instant gum, cereal bar, apple, carrot, sultanas, walnuts. Participants approached the stations one at a time, and at each one there was an instructor who helped the children to fill in a self-administered questionnaire created for the evaluation. The facilitator asked the question pointing to the selected product “Is this product healthy and why do you think so?” and wrote the answers on an answer sheet. At the end of the activity (after module 4), the students did the same task. Evaluation of the effectiveness of the education was possible by comparing the number of products indicated correctly in the initial part and the checking part. The answers to the pre- and post-test knowledge questions were evaluated. One point was awarded for each correct answer, while incorrect answers were not scored. A maximum of 14 points could be obtained. Based on the results, the following levels of nutritional knowledge were identified:−0–40% low level of knowledge−40–70% average knowledge level−70–100% high level of knowledge

### 2.5. Statistical Analysis

Results were collected using Microsoft Excel 2010 (Microsoft, Redmond, WA, USA), while statistical analysis was performed using Statistica 13.0 (TIBCO Statistica™, Paolo Alto, CA, USA). The results were analysed jointly before and after the project by gender. Analogously, the increase in nutritional knowledge was assessed. As a first step, it was checked whether the quantitative variables met the assumption of a normal distribution by performing the Shapiro-Wilk W test and normality plots. The Chi-square test was used to assess correlations. On the other hand, the Wilcoxon paired rank-order test was used to assess differences between the percentage of correct answers before and after the educational programme. For all analyses, a value of *p* < 0.05 was taken as statistically significant.

## 3. Results

The results of the project evaluation are presented in Table 3, Table 4, Table 5 and Table 6 and Figure 2.

Table 3 shows the knowledge scores of the surveyed students before and after the total educational project. Among the studied group, according to the adopted classification, 57.72% (86) of girls and 51.78% (87) of boys had a low level of knowledge. A medium level of knowledge was demonstrated by 29.53% of girls (44), and 35.12% of boys (59). A high level of knowledge was assessed in 12.75% of girls (19) and 13.10% of boys (22). After the nutrition education in both study groups, the level of knowledge increased; among the girls 99.33% (148) and among the boys 98.8% (166) had a high level of knowledge. Only one child still had a low level of knowledge and one girl and one boy each had a medium level. Table 4 shows the knowledge gain of the surveyed students after the educational programme. The greatest increase in knowledge was rated as high by 48.99% of girls (73) and 44.46% of boys (75). In the entire study group, the knowledge gain was rated as high to the greatest extent (46.69%). Low knowledge gain was in the smallest group of 16.4% of the children surveyed (Figure 2).

In Table 5, we present the percentage of correct answers when testing participants’ knowledge. Before the workshops, the median percentage of correct answers was 28.57% in both study groups. After education, the median increased to 100.00% also in both study groups. The knowledge gain among girls was 60.00% (median) and among boys 57.14% (median).

The results of the non-parametric Wilcoxon paired t-test allow us to conclude that there was a statistically significant difference between the percentage of correct answers given by the students before and after the nutritional education. This allows us to conclude that carrying out the educational programme resulted in an increase in nutritional knowledge (Table 6).

## 4. Discussion

The culinary–educational workshop programme presented in this paper aimed to increase interest in the topic of proper nutrition and healthy lifestyles, including improving the potentially unfavourable eating habits of participating children, which in the long term may reduce the prevalence of overweight and obesity in this group. As mentioned in the introduction, data from several studies indicate that it is possible to change eating habits to more health-promoting ones by increasing knowledge about proper nutrition. This can be achieved, among other things, by implementing appropriately designed nutrition education programmes [12,13]. Such an approach is particularly important for young children because at this age habits and food preferences are formed that often remain unchanged throughout life [14].

Based on the results of the author’s test, it was shown that the level of knowledge of the participants increased strongly in the short term. Before the nutritional education programme, the level of nutritional knowledge was mostly low or medium, whereas after the workshop series almost all respondents had a high level of nutritional knowledge. However, given that a child’s diet is highly influenced not only by acquired nutritional knowledge but also by patterns passed on from the family, this may explain why many school-based programmes targeting children of this age do not have satisfactory long-term results [18,29,30,31,32].

Too short a duration of the programmes implemented has been cited as one of the reasons for this. Guerra et al. [32] noted in their study that programmes longer than six months and those that included the children’s parents and caregivers were the most effective. Abruzzi et al. [33], in a meta-analysis of randomised clinical trials involving children aged 6–12 years, only included programmes that lasted longer than six months and showed an effect on reducing childhood obesity, but no effect on preventive management. Brave et al. observed no effect of health and nutrition education on body mass index (BMI) or the incidence of overweight and obesity but showed an effect on improving physical activity among a study group of Indian children [21].

A meta-analysis by Sobol-Goldberg et al. [34] showed a significant but small effect of educational programmes on reducing BMI in school-aged children, but not in adolescents. However, the best results were obtained if parents also participated in the programme and it lasted from 1 to 4 years. Such observations are in line with those of Polish educators, namely that children only pay attention to the products they eat for one to two weeks after a class on proper nutrition [35]. On the other hand, the need to involve parents in educational programmes has also been pointed out by other authors [36,37]. Relating the above considerations to the results of our research and the work of other authors, it can be assumed that the effectiveness of the undertaken educational activities may be influenced by the form of the conducted classes [27,33,38] consisting of various methods and teaching techniques [27]. Methods based on verbal communication, such as a lecture, are dominant in currently implemented programmes [27,33,38]. However, these methods are not very effective, as the essence of these methods is the transmission of ready-made information, which significantly limits the student’s participation in the learning process [27].

The innovation of the cooking and educational workshops described in the paper was the predominance of practical exercises. Even the typically theoretical content was conveyed to pupils in the form of activating activities. This was facilitated by the didactic aids used: the original didactic game “Go for Health”, props for the game “Familiada”, and food products—mainly vegetables from different groups and in different colours. During the activities in the cooking block, pupils independently prepared attractive meals and dishes from health-promoting ingredients.

A characteristic feature of the activities was that participants were encouraged to take an active part in the tasks, which has a positive effect on the student’s motivation and perceptual abilities [39,40]. Activation methods are particularly recommended in the teaching of preschool and early school-age children. An important and attractive form of physical activity for children at this age is play, undertaken in a variety of places, e.g., on the playing field, in the playground, at school, in training rooms, etc. The use of activating methods makes it possible to combine education and play, increasing the attractiveness of the teaching process and its effectiveness [40,41,42,43].

It is important to emphasise that effectively carried out nutrition education has both health and economic effects; moreover, it represents a long-term investment in the health of the population, which is particularly important in the child population. It is important to disseminate it, bearing in mind, however, that the formation of correct and sustainable dietary habits requires the cooperation of many professionals [44].

Although the number of nutrition education programmes is steadily increasing, few studies assess the actual impact of these programmes on specific dietary choices, especially among children [44,45,46]. According to many authors, the effectiveness of educational programmes should be carried out by objective methods that take into account not only the knowledge gained by those participating in the training but the actual change in dietary practices, as the links between higher nutritional knowledge and dietary behaviours and practices are ambiguous [18,47].

## 5. Strengths and Weaknesses of the Study (Project)

The strengths of our project and its evaluation are the participation of all grade 2 children in the Siemianowice Śląskie primary school and the multifaceted way of educating in terms of diversity. Due to the age of the children, we used the child interview method as a way of examining the increase in knowledge. For this purpose, the interviewers (i.e., the authors of the project) filled in the questionnaires before and after the education. In the case of young children, the first contact with the interviewer could be stressful, hence the answers may not be fully in line with the child’s knowledge and this is one of the weaknesses of our project. The child may have been shy and embarrassed. However, the presence of familiar teachers may have offset this stress created in the children. In the post-project surveys, the children no longer felt embarrassed during the evaluation.

## 6. Conclusions

The work presented here demonstrates the potential of nutritional education delivered in the form of cooking–education workshops to increase nutritional knowledge, which can be a tool to improve habits and eliminate dietary errors in the study population. However, future research is needed to verify whether such cooking–education workshops can lead to beneficial and long-lasting changes in eating habits over a longer period.

## Figures and Tables

**Figure 1 ijerph-19-13307-f001:**
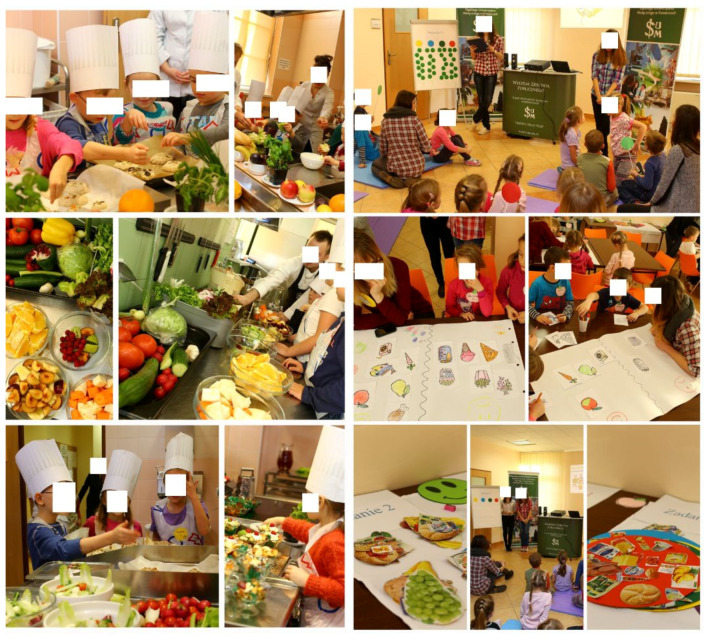
Educational workshops (MODULE 1) and cooking workshops (MODULE 3). (Photos by A.Białek-Dratwa).

**Figure 2 ijerph-19-13307-f002:**
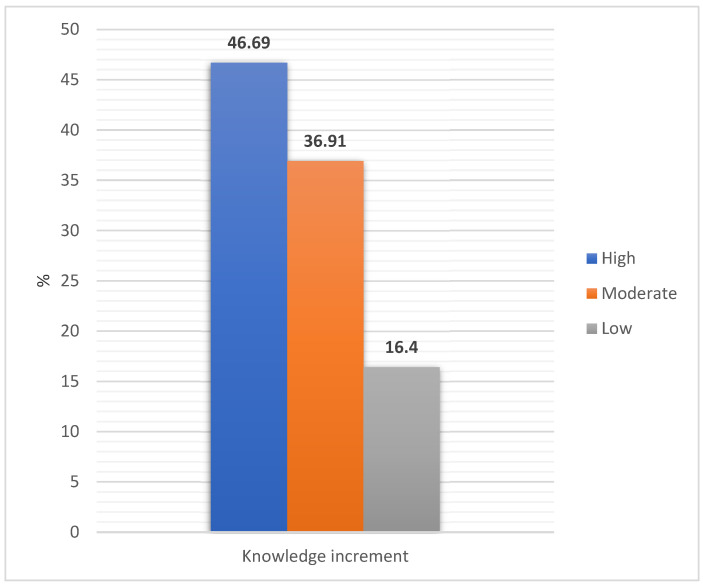
Increase in knowledge of surveyed students after the educational programme (chi-square test; *p* = 0.025, Cramer’s V = 0.15).

**Table 1 ijerph-19-13307-t001:** Overview of the implementation of week 1 of the “Colourful means healthy” project workshop.

WEEK 1 OF THE “COLOURFUL MEANS HEALTHY” PROJECT WORKSHOP		
	Educational Workshops (MODULE 1)	Sensory Workshops (MODULE 2)
**Purpose of educational workshops**	Getting pupils interested in the subject of good nutrition, through activities aimed at their cognitive development; shaping or changing (if inappropriate) attitudes, beliefs and opinions; developing practical skills; developing communication skills.	Making pupils aware of the role of the different senses in the evaluation of food products and familiarising participants with new tastes and smells, thereby promoting the acceptance of new or hitherto disliked food products. In addition, to develop practical skills in being able to diversify and compose tasty and healthy dishes and meals.
**As a result of the workshop, students will be able to**	− list the different levels of the Healthy Eating and Physical Activity Pyramid and the foods needed to grow and stay healthy, paying attention to limiting salt and sugar; − distinguish between products rich in nutrients and those of low nutritional value, and between processed and natural products; − communicate and thus cooperate in a group.	− identify the role of the different senses in evaluating food products; − plan varied, tasty and healthy dishes and meals.
**Ongoing educational workshops**	− The Healthy Eating and Physical Activity Pyramid for children and young people was presented; students were introduced to the different levels of the Pyramid, including the products recommended and contraindicated in daily nutrition. − The harmfulness of salt, the need to replace it with natural spices and the need to hydrate the body were also highlighted. − Participants were asked to solve crosswords and puzzles on such topics as rare fruit and vegetables or good nutrition. − An original educational game on the principles of good nutrition was played.	− The role played by the different senses in evaluating food products was presented. Pupils participating in the activity learnt what the sense organs are and the role they play in the evaluation of food products. − The ability to recognise basic tastes, odours and colour intensity was assessed. In addition, individual taste sensitivity (taste sensitivity threshold) was assessed and an odour recognition test was carried out. − The principles of food quality assessment were presented using the example of yoghurt, using the serialisation method.

**Table 2 ijerph-19-13307-t002:** Discussion of how week 2 of the “Colourful means healthy” project workshop was implemented.

WEEK 2 OF THE “COLOURFUL MEANS HEALTHY” PROJECT WORKSHOP		
	Cooking Workshops (MODULE 3)	Workshop on Molecular Gastronomy (MODULE 4)
**Purpose of educational workshops**	Getting pupils interested in good nutrition by encouraging them to choose quality food and to prepare their own meals.	Awakening children’s scientific curiosity and creative thinking by demonstrating the fast and spectacular process of freezing with liquid nitrogen.
**As a result of the workshop, students will be able to**	− recognise hitherto unfamiliar vegetables and herbs; − list the products with which salt and sugar can be substituted in dishes and were able to use them in the preparation of meals; − prepare colourful, tasty and healthy sandwiches themselves.	− Introduce an interesting application of molecular gastronomy in the preparation of healthy meals based on food products such as vegetables and fruits.
**During the implementation of the workshop**	− The principles of healthy eating were recalled, including the prevention of excessive salt and sugar consumption. − Students were introduced to modern equipment and facilities in the technology lab. − The principles of hygiene and safe working in the technology laboratory were presented. − A cooking class was held in groups, during which pupils prepared vegetable and fruit dishes without added sugar and salt, using substitutes.	− The physical properties of liquid nitrogen and the potential for its use in ultra-low temperature freezing were presented/presented. − The effects of very low temperatures on the texture of sample foods such as olive oil, fresh herbs and spinach were discussed, allowing the children’s imagination to be stimulated as to how they could be used as salad ingredients

**Table 3 ijerph-19-13307-t003:** Knowledge level of surveyed students before and after the educational programme.

Nutrition Education	Knowledge Level	Girls		Boys		Total	
		n	%	n	%	N	%
Before education	Low	86	57.72	87	51.79	173	54.57
	Medium	44	29.53	59	35.12	103	32.49
	High	19	12.75	22	13.10	41	12.93
After education	Low	0	0.00	1	0.60	1	0.32
	Medium	1	0.67	1	0.60	2	0.63
	High	148	99.33	166	98.80	314	99.05

**Table 4 ijerph-19-13307-t004:** Increase in knowledge of surveyed students after educational programme.

Nutrition Education	Knowledge Level	Girls		Boys		Total	
		n	%	n	%	N	%
Incremental knowledge	Low	25	16.78	27	16.07	52	16.40
	Medium	51	34.23	66	39.29	117	36.91
	High	73	48.99	75	44.64	148	46.69

**Table 5 ijerph-19-13307-t005:** Percentage of correct answers to knowledge questions.

Nutrition Education	Gender	Median	Minimum	Maximum	Quartile Range
Before education	Girls	28.57	0.00	87.50	28.57–42.86
	Boys	28.57	0.00	86.71	21.42–42.86
Total		28.57	0.00	87.50	28.57–42.86
After education	Girls	100.00	57.14	100.00	100.00–100.00
	Boys	100.00	28.57	100.00	100.00–100.00
Total		100.00	28.57	100.00	100.00–100.00
Incremental knowledge	Girls	60.00	12.50	100.00	42.86–71.43
	Boys	57.14	7.14	100.00	42.86–71.43
Total		57.14	7.14	100.00	42.86–71.43

**Table 6 ijerph-19-13307-t006:** Results of a statistical test assessing the difference between the percentage of correct answers before and after the educational programme.

Feature		Wilcoxon Paired *t*-test Result
Gender	Girls	*p* = 0.00
	Boys	*p* = 0.00
Programme editing educational	First	*p* = 0.00
	Second	*p* = 0.00

## Data Availability

The data presented in this study are available on request from the corresponding author. The data are not publicly available due to restrictions that apply to the availability of these data.

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
