# Peer review of "The Project “Colourful Means Healthy” as an Educational Measure for the Prevention of Diet-Related Diseases: Investigating the Impact of Nutrition Education for School-Aged Children on Their Nutritional Knowledge"

_ijerph, 2022, doi:10.3390/ijerph192013307_

Round 1

Reviewer 1 Report

The authors describe an educational nutrition program carried out among Polish school-aged children and analyse whether the program positively impacts the children’s nutrition knowledge. They find a statistically significant difference between the share of correct answers provided by the pupils before and after participating in the educational program.

The article is relevant and of interest to the journal’s readers and other stakeholders. It is straightforward and well structured. Methods are appropriate and are adequately described. The discussion and conclusions are interesting. However, revisions are needed along several aspects:

Broad comments:

·       Even though the project’s aim has been extensively explained in the introduction, I miss an outline of the aim of the paper. Though related, it should be different.  

·       I would like to know why the fourth module is included. I think it is a good idea, but I think it would be good to include it in the text to make it clear to the reader.

·       Results need to be presented in more detail. The authors only comment Table 6. I agree on the fact that Table 6 shows the key results of the paper, but Tables 3-5 should also be commented. Moreover, I do not see clearly what Table 5 shows.

Specific comments:

·       Abstract: Just the same as the authors have included sections for “results” and “Conclusion”, I would suggest including sections for objectives or methods.     

·       Lines 98-99: Is it “colorful” of colorfully? I would suggest being consistent throughout the paper.

·       Line 142: this is not a thesis but an article. I would suggest rewriting the sentence.

·       Line 277: A full stop is missing.

·       Line 286: Figure 1 is not correctly numbered.

·       Tables 3-6: I would suggest removing the heading “before and after nutrition education” in the first row and column of the tables.

·       Table 3: The number “51.78” corresponding to boys with low knowledge level before education is not correctly rounded. The number “32.50” corresponding to total with medium knowledge level before education is not correctly rounded.

·       Line 334: BMI has not been specified first by its full name.

·       Line 347: The text reads “such as a lecture or a lecture”. I would suggest correcting the sentence.

·       Lines 362-363: The text reads “This is because an important and attractive form of activity for children at this age plays, undertaken in a variety of settings”. I cannot understand the sentence and I would suggest rewriting it.

·       Figures are not correctly numbered.

Author Response

Dear Reviewer

Thank you very much for your time and thorough review of our article 'The project "Colorful means healthy" as an educational measure for the prevention of diet-related diseases - investigating the impact of nutrition education for school-aged children on their nutritional knowledge'.
Below, we answer all your questions and suggestions in turn. In the text, we have highlighted all the changes in yellow so that you can easily follow the changes we have made. We hope you find our explanations clear and easy to read. Should you need to be more specific in your answers, please let us know. 

Comments 
1. we understand your comment. We actually described the purpose of our project and not of the article. We have added the purpose of the publication.
2. the fourth module was included to get children to consolidate certain eating behaviours combined with 'scientific magic'. Children aged 7-9 remember, they learn through experience, curiosity. Therefore, we used molecular gastronomy as 'scientific magic' to make the children curious and fix in their minds. In module 4, using liquid nitrogen and other molecular gastronomy techniques, we made healthy dishes together with the children, e.g. using liquid nitrogen we created ice cream based on fruit alone in front of the children's eyes. We used Fermentation , calcium added to juice and an alginate drip. to create caviar from chokeberry juice, and agar and special shaping tubes to create spaghetti from carrot juice. We described this module in more detail in the methodology. 
3) We have described Tables 3 to 5 in more detail. We hope that it is now easier to understand.
4. in table 5 we have presented the percentage of the number of correct answers checking knowledge we have presented the results using min, max, median and quartile range values to show the level of knowledge among the children. The average value would be a misleading value for the recipient of this article.
(5) In the executive summary, we added an objective. However, due to the word limit in the executive summary, we were unable to briefly describe the methods used in the project.
6 We have consistently changed throughout the article.
7) We removed the word thesis replacing it with "article".
8) We removed the ".".
9) We have renumbered the figure.
10) In Tables 3-5, we removed the word before and after from the heading.
11. we have rounded the numbers correctly. We apologised for our error.
12 We have corrected.
13. we have paraphrased this sentence, we hope it is now understandable. We apologise for our error.
14) We have corrected the figure numbers.
Thank you again for your time and all your comments, making our article better understood and we hope.

With best regards 
Agnieszka Białek-Dratwa

Reviewer 2 Report

In the first place, I'd like to congratulate the authors on their study; it's an interesting paper that provides valuable information about the study of obesity and overweight in the children population.

Regarding the comments or suggestions on the article, I let you know my observations. I think that the abstract and key words, as well as the introduction, are accurate and relevant. The materials and methods section presents detailed and pertinent information. However, I would like you to include the latest Healthy Eating and Physical Activity Pyramid for children and young people in Poland as supplementary material (mentioned as part of module 1), since as a researcher from Latin America I haven't been able to find it on the internet. I'd also like you to specify whether the workshops are based on the Polish food culture and/or local diet, or on general healthy eating. Results section:  In addition, I'd like to know if the authors have qualitative data such as the children's food preferences (regarding flavors, do they prefer sweet, sour, salty flavors?). Even though this is an education intervention, if anthropometric measurements were taken in children at the beginning and end of the intervention, it would be of interest to have those data. In the Conclusions section, I'd like you to elaborate on how these type of interventions could be incorporated into public policies.

Comment on spelling: while there are no evident misspellings, I noticed that that the word "colorful" is written in both British and American English throughout the text, only one spelling should be used to maintain consistency.

The name of the project is also written in two different ways: "Colourfully means healthily" (once, line 77) and "Colorful means healthy".

Author Response

Dear Reviewer, 

Thank you very much for reviewing our work and your time to make our publication better. We have highlighted all the changes we have made in response to both reviewers' reviews in yellow to make it easier to follow the changes.

Below we respond to your suggestions in turn.
1 We have legal doubts whether we can use the Healthy Eating and Physical Activity Pyramid for children and young people in Poland in the publication. Because on the website of this authority the following clause appears "*The Pyramid of Healthy Eating and Physical Activity for Children and Young People, including all its text and graphic elements ("Pyramid") is the property of the Dr. med. A. Szczygiel Institute of Food and Nutrition in Warsaw and is subject to legal protection under copyright law. Any use of the Pyramid other than for personal use, and in particular the reproduction, distribution, multiplication and modification of the Pyramid requires the prior written consent of the Institute of Food and Nutrition. The use of the Pyramid without the appropriate permission may be considered as an act that infringes the legally protected interests of the Institute of Food and Nutrition." However, we have included a link to this food pyramid in the references so that it is available to audiences around the world.
2 We clarified in the text that the cooking workshop was based on the general principles of healthy eating, which include the 10 principles:
- Eat 5 meals regularly and remember to drink water frequently and brush your teeth after eating.
- Eat a variety of fruit and vegetables as often and as much as possible.
- Eat cereal products, especially whole grains.
- Drink at least 3-4 glasses of milk a day (you can replace it with natural yoghurt, kefir and - partly - cheese).
- Eat lean meat, fish, eggs, pulses and choose vegetable fats over animal fats.
- Avoid sugary drinks and sweets (replace them with fruit and nuts).
- Don't add salt to your food, eat salty snacks or fast food.
- Be physically active at least one hour a day (limit TV watching, computer use and other electronic devices to 2 hours).
- Get enough sleep so your brain can rest.
- Check your height and weight regularly.
We have added information about this in the text.
3 In this study, unfortunately, we did not investigate the sensorium of taste among children. However, in subsequent projects that we carried out in our Department of Dietetics, we have taken away just such studies to assess taste preference. We regret very much that we were not able to do this in this project, because the results in the subsequent studies were very interesting. However, we had to bear in mind that workshops for such an age group cannot be too long and engaging, because children aged 7-9 will not be able to concentrate properly for the whole time. Also in this study, we did not undertake a body composition analysis before and after the project, as the aim of the project was not a nutritional intervention and weight loss process for children. At the same time, the time between one stage and the other was too short to make an assessment. However, we think it would be an interesting idea to survey this group again in terms of nutritional knowledge. And to evaluate the long-term effects. Thank you for the idea of further research.
4 Actually, the word colourfully and colour does appear, so we have changed and standardised the spelling to colourful and colour throughout the text. We apologise for our errors.

Thank you again for your review and your time.

Our warmest greetings
Agnieszka Białek-Dratwa

Round 2

Reviewer 1 Report

The authors' replies have addressed my comments quite satisfactorily. The article has been improved significantly. I just have a couple of very minor comments:

·       Regarding the abstract, I suggested including sections for objectives or methods. I meant including the words too (“Objective:” and “Methods:”), just as the authors do for “Results:” and “Conclusions:”.  

·       Line 306: A full stop is still missing. 

Author Response

Dear Reviewer.

We have added the division and the words "Objective:" and "Methods:".
And we apologise for not noticing the absence of the punctuation mark "."

We hope that everything is now ok.
Thank you for your time.

Best regards
Agnieszka Białek-Dratwa
